# Anhydrous interfacial polymerization of sub-1 Å sieving polyamide membrane

Guangjin Zhao[1], Haiqi Gao[2], Zhou Qu[1], Hongwei Fan ®[1] ✉ & Hong Meng ®[2] ✉

Highly permeable polyamide (PA) membrane capable of precise ionic sieving can be utilized for many energy-efficient chemical separations. To fulfill this target, it is crucial to innovate membrane-forming process to induce a narrow pore-size distribution. Herein, we report an anhydrous interfacial polymerization (AIP) at a solid-liquid interface where the amine layer sublimated is in direct contact with the alkane containing acyl chlorides. In such a heterophase interface, water-caused side reactions are eliminated, and the amines in compact arrangement enable an intensive and orderly IP reaction, leading to a unique PA layer with an ionic sieving accuracy of 0.5 Å. The AIP-PA membrane demonstrates excellent separation selectivities of monovalent and divalent cations such as $Mg^{2+}/Li^+$ (78.3) and anions such as $Cl^-/SO_4^{2-}$ (29.2) together with a high water flux up to $13.6\,L\,m^{-2}\,h^{-1}\,bar^{-1}$. Our AIP strategy may provide inspirations for engineering high-precision PA membranes available in various advanced separations.

Precise separation of ions with similar size is a fundamental and challenging step in the chemical industry[1–4]. Polyamide membrane (PA) has been successfully used in reverse osmosis desalination and nanofiltration for separation of monovalent and divalent anions (i.e., $Cl^-/SO_4^{2-}$) through a combination of electrostatic repulsion (Donnan effect) and size sieving[5–12]. However, there is often a dilemma faced by the existing PA membranes for selective separation of similarly sized metallic cations such as $Mg^{2+}/Li^+$ [13]. Despite the typical feature of charge repulsion, the construction of desirable PA membrane with highly uniform pore sizes is urgently required to resolve this issue[14,15]. Therefore, it is essential to rationally regulate the membrane-forming process to generate a precise ion sieving and solute differentiation, which is technically difficult due to the intrinsic amorphous structure of PA and the existence of side reactions[16,17].

Generally, the PA selective layer is constructed by interfacial polymerization (IP), where the amine and acyl chloride that are, respectively, dissolved in water and alkane react with each other at the liquid-liquid interface[18]. This process involves a counter diffusion of the two reactive monomers, and the condensation reaction at the water-alkane interface is extremely fast and uncontrolled[19], often resulting in a wide pore-size distribution in the obtained PA layers[20,21].

To tackle this challenge, a comprehensive understanding of the IP process is needed in order to tune the homogeneity of membrane pores from a mechanistic perspective. Liang et al. proposed an improved IP of introducing surfactants at the oil-water interface to promote the trans-interfacial diffusion behavior of amine monomers, and the formed PA membrane has a relatively high homogeneousness of structure[13]. Shen et al. explored an inorganic salt-mediated IP to regulate the nanoscale homogeneity of PA-based thin film composite (TFC) membranes[22]. Nevertheless, these processes are still water-bearing, which suffer the water-caused side reactions.

In view of the above analysis, in this study, we present an anhydrous IP (AIP) to prepare highly permeable PA membranes suited for selective ionic separation. This process was conducted by sublimating the amine onto a porous substrate as the solid phase to react with the liquid phase of acyl chloride alkane solution at the solid-liquid interface. Due to the absence of water, side reactions of acyl chloride hydrolysis were completely eliminated, and simultaneously the amine monomers in compact arrangement enabled the condensation reaction to be more intensive and ordered, thereby achieving the sub-1 Å sieving PA membranes. Apart from affording a kind of high-precise ionic separation membrane for potential applications including

[1]College of Chemical Engineering, Beijing University of Chemical Technology, Beijing 100029, PR China. [2]State Key Laboratory of Chemistry and Utilization of Carbon Based Energy Resources, College of Chemistry, Xinjiang University, Urumqi 830046, PR China. ✉ e-mail: fanhongwei@mail.buct.edu.cn; menghong@xju.edu.cn

lithium extraction and rare-earth recycling, the unique AIP concept will also bring implications in the construction and structural regulation of PA-based membranes.

## Results

### Preparation of AIP-PA membrane

For the conventional IP (CIP) of PA membrane (Fig. 1a, Supplementary Fig. 1), there is a counter diffusion of piperazine (PIP) and trimesoyl chloride (TMC) at the water-n-hexane interface, and due to the much higher solubility of the former in n-hexane than that of the latter in water, the condensation reaction occurs in the organic phase near the interface[23]. During this process, on the one hand, TMC hydrolysis would produce noncross-linkable sites, inevitably leading to the wide pore-size distribution and even the potential defects in the resulting PA layers[17]. On the other hand, the uncontrollable reaction as a result of the random counter diffusion of reactive monomers often causes an inhomogeneous CIP-PA membrane with a high void ratio[20–22]. Obviously, how to create an anhydrous reaction condition and regulate the controlling steps of interphase mass transfer of amine monomers is the key to achieve the narrow pore-size distributed PA membrane.

Figure 1b and Supplementary Fig. 2 show the AIP process which is performed in such an expected way, including two steps. Firstly, PIP molecules were volatilized and adsorbed onto the surface of a polyacrylonitrile (PAN) ultrafiltration substrate to form a solid-phase layer. The PIP content at the interface could be controlled by altering the volatilization temperature and time. Secondly, the AIP was carried out by immersing the PIP-containing PAN substrate into the TMC n-hexane solution. This anhydrous interface can entirely avoid the hydrolysis of TMC and the subsequent side reaction. Moreover, it could be reasonably speculated that the amine monomer molecules with no diffusion in the aqueous phase directly reacted with the TMC in the n-hexane at the solid-liquid interface, which facilitated the IP reaction

in an intensive and ordered manner. After an interval, the AIP-PA membrane was obtained by washing with deionized water.

### Morphological and structural analysis of AIP-PA membrane

Top-view Scanning Electron Microscopy (SEM) images (Fig. 2a) revealed similar surface morphologies of the CIP-PA membrane and AIP-PA membrane, which are both dense and consist of granular protrusions. The surface roughness of the AIP-PA membrane is 7.2 nm relatively higher than that (6.82 nm) of the CIP-PA membrane measured from Atomic Force Microscopy (AFM) (Fig. 2b, c). This result is probably due to that the promoted reaction occurring during AIP released more heat, giving rise to more escaped gases dissolving in the alkane phase[24]. According to the Windsor theory, the higher surface roughness will reduce the value of the water contact angle (WCA) and increase the hydrophilicity (Supplementary Fig. 3). From Fig. 2d of Transmission Electron Microscopy (TEM) images, the AIP-PA selective layer has a thinner thickness of about 35 nm as compared to that of the counterpart (CIP-PA selective layer). This could be ascribed to a more remarkable self-limiting phenomenon existing in the reaction-diffusion process of AIP that is out of thermodynamic equilibrium[10,25]. The thinner AIP-PA selective layer will have a smaller transport resistance, in favor of enhancing water permeance. The chemical composition and structure of the PA membranes were analyzed by Attenuated Total Reflection-Flourier Transformed Infrared Spectroscopy (ATR-FTIR) and X-ray photoelectron spectroscopy (XPS). As shown in Supplementary Fig. 4, the amide group formed from the reaction of solid PIP and TMC in n-hexane is responsible for the new stretching vibration peak of C = O bond located at 1630 cm$^{-1}$, confirming the successful synthesis of the PA layer by AIP. The full XPS survey spectra (Fig. 2e, Supplementary Table 1) indicate the presence of three characteristic peaks of C (C1s), N (N1s) and O (O1s) for both AIP-PA and CIP-PA selective layers[26]. Notably, by comparing the high-resolution spectra of the deconvoluted C1s, O1s, and N1s, the N-C = O

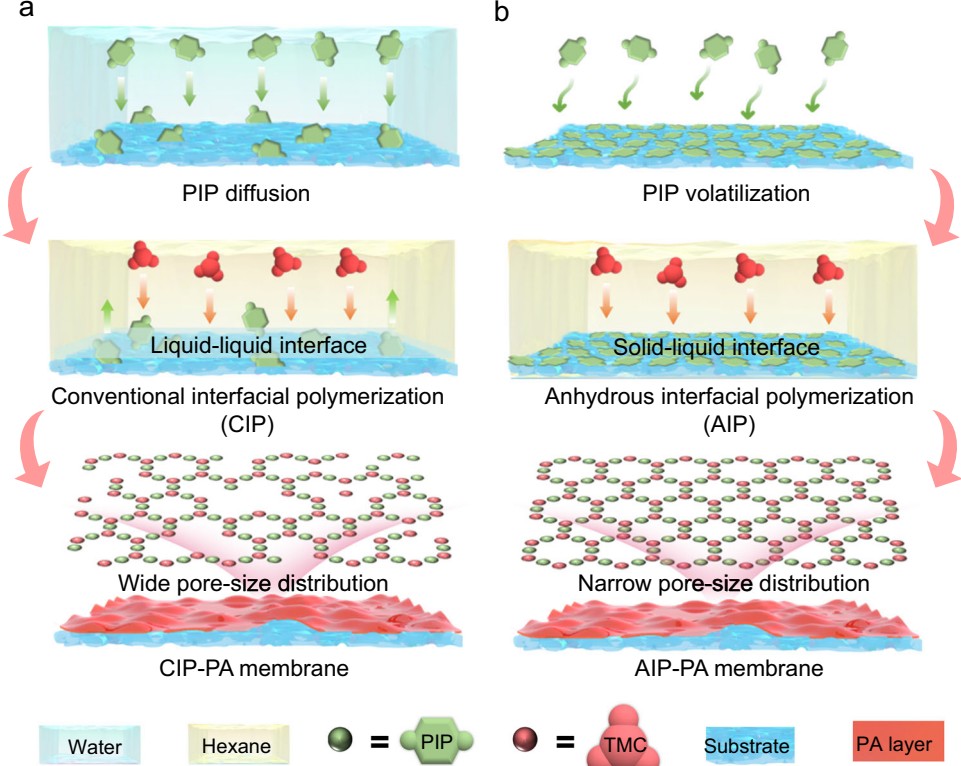

**Fig. 1 | Schematic illustration of CIP and AIP processes.** Scheme depicting the preparation of (**a**) CIP-PA and (**b**) AIP-PA membranes, where PIP, TMC and PA are abbreviations for piperazine, trimesoyl chloride and polyamide, respectively.

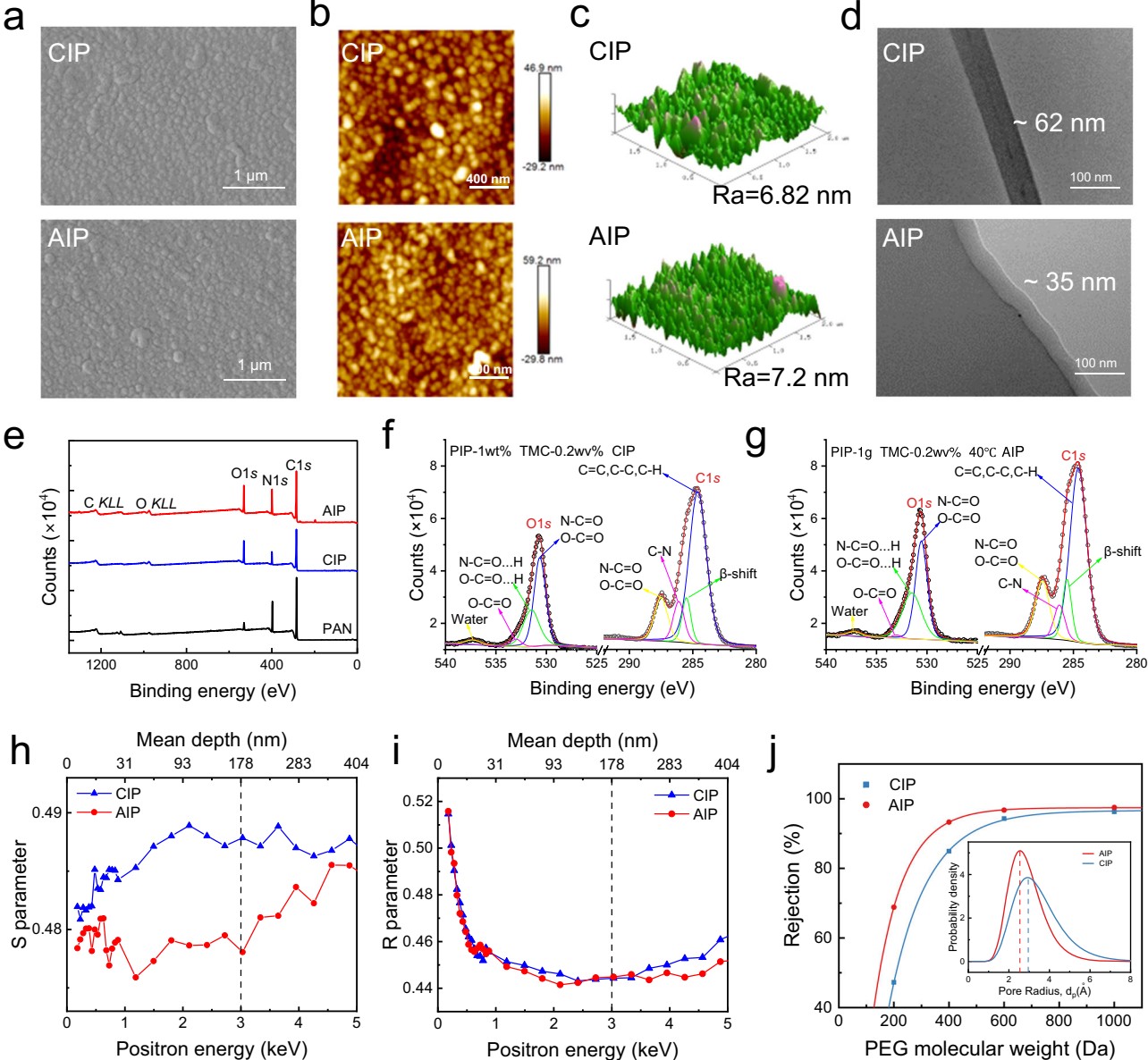

**Fig. 2 | Morphological and structural analysis of AIP-PA and CIP-PA membranes. a** Top-view SEM images. **b** 2D and (**c**) 3D AFM images. **d** Cross-sectional TEM images. **e** Full XPS survey spectra. High-resolution spectra of deconvoluted (**f**) C1s and (**g**) O1s. **h** S parameter and (**i**) R parameter of the positron incident energy.

**j** Rejection of PEG with different molecular weight by PA membrane obtained by AIP (red dots and curve) and CIP (blue squares and curve). Inset: pore-size distribution derived from rejection curves of PEG.

content in AIP-PA is evidently higher, while the O-C = O content is lower (Fig. 2f, g, Supplementary Fig. 5). Moreover, the surface zeta potential shows that the AIP-PA membrane has less negative charges (from side reactions) than that of the CIP-PA membrane at different pH values (Supplementary Fig. 6). These results indicate that the AIP could effectively restrain the TMC hydrolysis, facilitate the condensation reaction and increase the compactness of the resulting PA layer.

The evolution of free volume and average pore-size distribution of the AIP-PA and CIP-PA membranes were examined by positron annihilation spectroscopy (PAS) (Fig. 2h, i) and rejection experiment of neutral solutes of polyethylene glycol (PEG) (Fig. 2j). Generally, in the quantitative PAS data, the ordinate S parameter represents the relative value of the free volume depth distribution in the polymer system, and the ordinate R parameter manifests the details on the occurrence of large pores (nm to μm)[27,28]. It follows from Fig. 2h that the S-parameters of AIP-PA membrane fluctuated around the value of 0.48 within the range of 3 keV of positron energy (a typical testing scope for detecting

PA layer), whereas a gradual upward trend was observed for the CIP-PA membrane. Moreover, the S-value of the former is always lower than that of the latter across the whole detection depth. This measurement suggests the more uniform structure with a smaller and free volume in the AIP-PA membrane. Despite the similar variation trend of R-parameters (Fig. 2i), the R-value of AIP-PA membrane shows a more significant decline with the increase of positron energy and also has a smaller bottom point. Based on the previous studies[27,28], the difference value of the depth corresponding to the initial point and the bottom point of the R-parameter depicts the thickness of the PA selective layer, and the minimal R-value represents the minimum pore strength. These differential results of R-parameters also imply that the AIP-PA membrane has a lower pore strength and thinner thickness which is consistent with the experimental data. Figure 2j shows that the measured molecular weight cut-off (MWCO) of AIP-PA membrane is visibly smaller than that of the CIP-PA membrane. A point worth noting is that the MWCO is slightly larger than those determined by the near-

spherical calibration substances (e.g., glycerol, glucose, sucrose, raffinose)[29–31]. Taking the molecular conformation into consideration that the chain-like PEG series are of more permeable, this is to be expected and understandable. Moreover, as compared to other reported PA membranes[32–34], our AIP-PA membrane has a relatively lower retention of PEG-200 (equivalent Stokes radius of 3.2 Å), but has a comparable retention of PEG-400 (equivalent Stokes radius of 4.7 Å). This could be another reason leading to the relatively larger value of MWCO obtained from the fitting curve. Whereas, this phenomenon just can reflect the structural uniqueness of the AIP-PA membrane that can precisely separate mono- and divalent cations. The low retention of PEG-200 denotes more low-valent ions (generally hydration radius less than 3.9 Å) could permeate through the membrane, while the similar retention of PEG-400 signifies that it could remain an efficient retention of high-valent ions (generally hydration radius larger than 4.0 Å), thereby resulting in a large retention difference and a precise ionic recognition. Nonetheless, under the same testing condition these findings further demonstrate the validity of the AIP strategy in terms of precisely regulating the pore structure[11,35,36]. Accordingly, the calculated pore-size distribution of the former is also more sharpened. Especially, the pore size of the AIP-PA membrane is smaller (about 5.08 Å) and is exactly located between the size of monovalent and divalent ions, indicating a potent molecular sieving for selective ionic separation.

In addition, the effect of solid PIP content on the AIP-PA selective layer structure was studied by altering the volatilization temperature. It was found that the volatilization rate of PIP increased from 4.59% to 8.92% with the increase in temperature from 30 °C to 50 °C (Supplementary Fig. 7), and the amount of PIP monomers deposited on the PAN substrate surface increases from 0.01 mg·cm$^{-2}$ to 0.67 mg·cm$^{-2}$ (Supplementary Figs. 8, 9). It can be seen from surface SEM images that the AIP-PA membrane prepared at 30 °C (named as AIP-PA@30 membrane) is not continuous and has visible defects. As the rising of the PIP volatilization temperature, the surface becomes denser and rougher (Supplementary Figs. 10–12), and also the thickness of the AIP-PA selective layer gradually increases (Supplementary Fig. 13). Correspondingly, the surface hydrophilicity and zeta potential values were enhanced (Supplementary Figs. 14, 15), due to the increased content of hydrophilic C-NH bonds on the membrane surface (Supplementary Fig. 16, Supplementary Table 2). Moreover, the AIP-PA@40 membrane shows the highest C-N content and the lowest -COOH content from the deconvoluted C1s and O1s, respectively (Supplementary Figs. 17, 18, Supplementary Tables 3, 4), revealing the best cross-linking characteristics. These results further suggest that the AIP process is closely associated with the quantity of amine monomers on the substrate surface, essentially differing from the CIP process during which the diffusion of amine monomers in the aqueous phase is the controlling step.

## Ion separation performance of AIP-PA membrane

Membrane performance for selective ion separation was measured through a cross-flow nanofiltration experiments (Supplementary Fig. 19). Before systematic evaluation, the effects of preparation parameters including volatilization temperature, volatilization time and solid mass of PIP, and the operation pressure on the performance of the resulting AIP-PA membrane were studied by using a 1000 ppm Na$_2$SO$_4$ solution as the feed. For example, with the increase of volatilization temperature from 30 °C to 50 °C, the rejection rate of Na$_2$SO$_4$ goes up first and then declines, but the water permeate flux just behaves in the opposite way (Supplementary Fig. 20). The change of salt rejection is closely related to the membrane structure, which has a positive correlation with the change of C-N content. The Na$_2$SO$_4$ rejection rate reaches a maximum value of 96.45% at 40 °C, suggesting that the volatilized PIP content at this temperature is enough to react with the TMC monomers to form a dense PA membrane. Combined

with the investigation results of volatilization time and solid mass of PIP (Supplementary Figs. 21, 22), the optimized condition is determined as: 40 °C of volatilization temperature, 10 min of volatilization time and 1 gram of PIP solid mass for volatilization, respectively. In addition, there is no obvious deterioration of the Na$_2$SO$_4$ rejection rate (above 94%) as the operation pressure was increased from 2 bar to 6 bar (Supplementary Fig. 23), and simultaneously, the water permeate flux gradually increases, though a little below that of the CIP-PA membrane, probably due to the denser AIP-PA selective layer with a smaller pore size. It is worth mentioning that the preparation condition of CIP-PA membrane was also investigated (Supplementary Fig. 24), and the optimized nanofiltration performance was used as the benchmark of AIP-PA membrane in the subsequent discussion.

It follows from Fig. 3a that the AIP-PA membrane can intercept various salts such as Na$_2$SO$_4$, MgSO$_4$, MgCl$_2$, CaCl$_2$, and their rejection rates are all above 90%, which are superior to that of the CIP-PA membrane. The valence ratio of the anionic (Z-) and cationic (Z+) species of the salt is often sensitive to the salt rejection according to Donnan exclusion theory, and the salt rejection is better at a larger valence ratio (Z-/Z+ for negatively charged membranes and Z+/Z- for positively charged membranes)[37,38]. Generally, the CIP-PA membrane surface has more negative charges, and therefore, a stronger repulsion is expected for Na$_2$SO$_4$ with a valence ratio of 2, whereas a weaker rejection is anticipated for MgCl$_2$ and CaCl$_2$ with larger ionic radii. Intriguingly, while the AIP-PA membrane has a reduced surface negative charge and even a positive charge (when pH=3 in Supplementary Fig. 6), it displays high rejections of Na$_2$SO$_4$, MgSO$_4$ and MgCl$_2$ as well as CaCl$_2$, but a low rejection of NaCl (<20%).

The excellent salt rejection and solute differentiation encourage us to examine the sieving accuracy of the AIP-PA membranes by intercepting different small solutes, and the molecular size of each solute was uniformly designated by Stokes radius ($r_s$) (Supplementary Table 5). As shown in Fig. 3b, for the AIP-PA membrane, there is a sharp cutoff between the Li$^+$ (2.4 Å) rejection and Ba$^{2+}$ (2.9 Å) rejection, and the sieving accuracy reaches 0.5 Å. This result indicates a sub-1 Å sieving property for selective cationic separation, benefiting from the refined AIP-PA layer structure with a narrower pore-size distribution as demonstrated in Fig. 2h–j. In contrast, the CIP-PA membrane has a sieving accuracy of only 1.2 Å. (between 2.5 Å and 3.7 Å) (Fig. 3c) for the cationic separation. Although the Donan effect of more negative charges from side reactions allows the CIP-PA membrane to resist multivalent anions such as sulfate, the repulsion to multivalent cations is weak and the rejection varies dramatically with the ionic size. It is worth noting that, besides the precise sieving of monovalent and divalent cations, the AIP-PA membrane also exhibits a retention of above 97% for GdCl$_3$, SmCl$_3$ and LaCl$_3$, which is promising to separate water-soluble rare earth salts.

Extracting lithium from salt-lake brine is of great significance for energy storage applications because of the abundant lithium resources in brines. Given the sub-1 Å sieving accuracy, the AIP-PA membrane is expected to have the potential in addressing the challenges of magnesium/lithium separation from salt-lake brines. Therefore, we further investigated the performance of the AIP-PA membranes for selective separation of magnesium and lithium from aqueous solution. First, the effect of MgCl$_2$ and LiCl concentrations on the membrane performance was examined, and the results are shown in Supplementary Figs. 25, 26. The water permeate flux reduced with the increase of MgCl$_2$ and LiCl concentrations from 1000 ppm to 5000 ppm, probably due to the increased osmotic pressure differential between the feed and the permeate solution[39]. In the meantime, at any concentration the AIP-PA membrane can intercept 90% of Mg$^{2+}$ while allowing at least 80% of Li to pass through. This phenomenon indicates that the Li$^+$ with a low hydration radius (0.38 nm) and hydration free energy (474 KJ/mol) are more permeable than that (0.43 nm, 1828 KJ/mol) of the Mg$^{2+}$ in the AIP-PA membrane, and the sub-1 Å size sieving played a

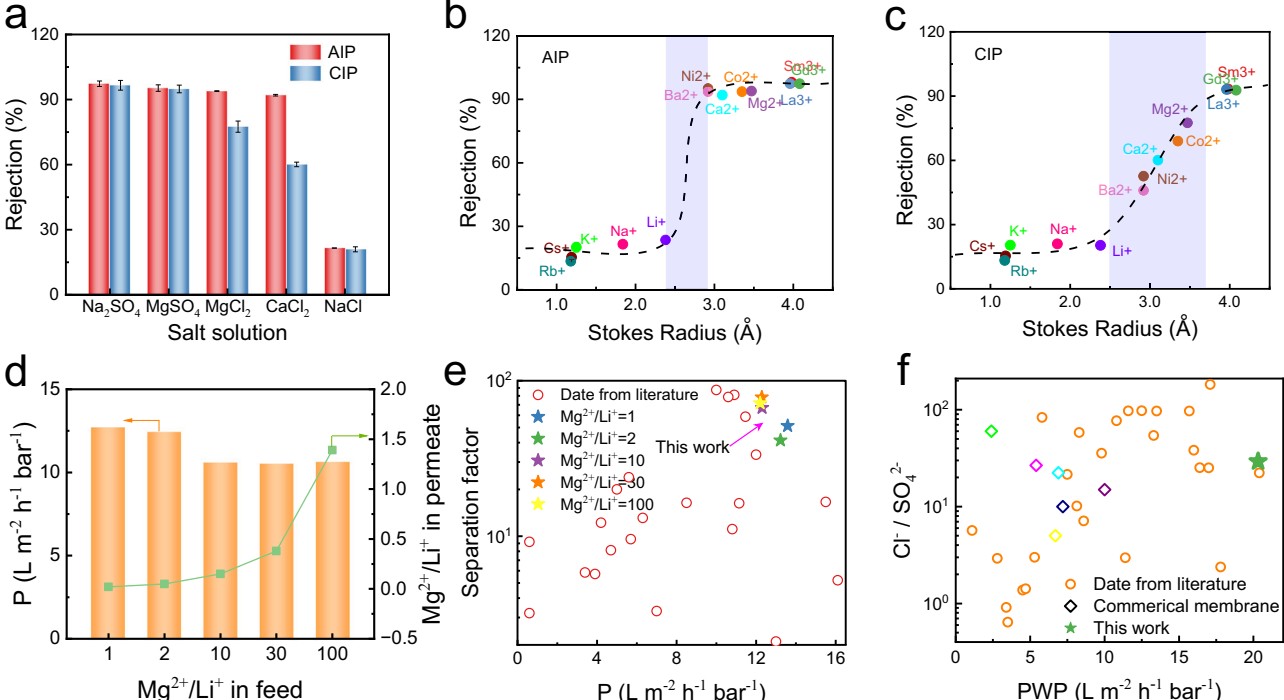

**Fig. 3 | Ion separation performance. a** Rejection of different salts by AIP-PA membrane and CIP-PA membrane (salt concentration: 1000 ppm; applied pressure: 4 bar). Rejection of different solutes as a function of the Stokes radius (for details see Supplementary Table 5) for (**b**) AIP-PA membrane and (**c**) CIP-PA membrane. **d** Comparison of Cl⁻/SO₄²⁻ separation performance of AIP-PA membrane and other PA membranes reported in literature. **e** Effect of $Mg^{2+}/Li^+$ ratio in the feed on the nanofiltration performance (salt concentration: 2000 ppm; applied pressure: 4 bar). **f** Comparison of $Mg^{2+}/Li^+$ separation performance of AIP-PA membrane and other PA membranes reported in literature. The error bars represent the standard deviation of data from three replicate measurements.

main role in $Mg^{2+}/Li^+$ separation, unlike other reported most nanofiltration membranes largely depending on the Donnan effect[40,41]. The significant interception differential between $Mg^{2+}/Li^+$ enables an effective removal of $Mg^{2+}$ from the combinations of $Mg^{2+}$ and $Li^+$ with different $Mg^{2+}/Li^+$ ratios (Fig. 3d). As the $Mg^{2+}/Li^+$ ratio in the feed increases even up to 100, the $Mg^{2+}/Li^+$ ratio in the permeate is still very low (below 1.5), and the water permeate flux remains above 12 L m⁻² h⁻¹ bar⁻¹. Specifically, for the $Mg^{2+}/Li^+$=30, the separation factor can reach 78, which is higher than most of the existing $Mg^{2+}/Li^+$ separation membranes. A comprehensive comparison in terms of $Mg^{2+}/Li^+$ separation factor and water permeate flux reveals a more excellent overall performance of our AIP-PA membranes at different $Mg^{2+}/Li^+$ ratios (Fig. 3e, Supplementary Table 6). Moreover, the AIP-PA membrane demonstrated a good stability regardless of the single MgCl₂ solution or the MgCl₂/LiCl mixed solution (Supplementary Figs. 27, 28). In addition, the separation factor of Cl⁻/SO₄²⁻ is as high as 29.27 and the comprehensive performance is comparable among various nanofiltration membranes, also suggesting the great potential of AIP-PA membrane in selective monovalent and divalent anions separation (Fig. 3f, Supplementary Table 7).

Additionally, introducing a gutter layer is an emerging approach to effectively tune the structure and properties of PA membrane by improving amine distribution at the interface and avoiding pore infiltration of amine[11,16]. We also incorporated a graphene oxide layer as the gutter layer to further regulate the AIP process (Supplementary Figs. 29, 30) in an attempt to obtain more compact and more refined pore structure in the resulting membrane (GL-PA membrane) (Supplementary Figs. 31–35), by making the distribution of sublimated PIP molecules on porous substrate more uniform and denser. As expected, Supplementary Fig. 36 displays a smaller MWCO of 300 Da and a smaller pore size (about 4.62 Å) than that of the AIP-PA membrane (Fig. 2j). Based on this, the GL-PA membrane showed higher salt rejections, despite a little decease in water permeance flux

(Supplementary Fig. 37). For example, the rejection for Na₂SO₄ and MgSO₄ can reach up to 97.92% and 98.60%, respectively, and the rejections for divalent cationic chloride salts were all above 94%. Moreover, the GL-PA membrane exhibited rejections of 35.06% and 30.89% for LiCl and NaCl, respectively, which were at least 9% higher than those of AIP-PA membrane. These results imply that the gutter layer enabled a tighter structure with narrower pores in the resulting GL-PA membrane, which was beneficial for selective rejection of solutes. A negative result of the preparation of PA layer on a more porous substrate of polyether sulfone (with a wide pore size of 0.22 μm) (Supplementary Fig. 38) also suggests the possible assistance of the gutter layer to the AIP strategy. It can not be denied that the current continuous production of AIP-PA membrane is challenging compared to the CIP process, but the AIP strategy still has the potential for scaling up. Further, we demonstrated the scalability by preparing larger-area membranes with an area of 314 cm² using an improved equipment (Supplementary Figs. 39–42), and sheets of such membranes were placed inside a custom-made membrane module for NF measurement (Supplementary Figs. 43, 44). This membrane module could retain MgSO₄ up to 95.88% together with a treatment capacity of 601.39 L m⁻² d⁻¹. Meanwhile, effective separation of $Mg^{2+}$ and $Li^+$ could also be realized due to the large difference of rejection (above 70%) for $Mg^{2+}$ and $Li^+$, demonstrating the potential of the AIP-PA membranes in industrial application (Supplementary Fig. 45).

## Discussion

To elucidate the $Mg^{2+}/Li^+$ separation mechanism, we further carried out molecular dynamics (MD) simulation to investigate the ion transport behavior through the AIP-PA membrane in thickness of 4 nm (Supplementary Table 8, Supplementary Figs. 46, 47). Figure 4a shows the radial distribution function (RDF) of ion-oxygen in bulk solution. Both $Mg^{2+}$ and $Li^+$ have two peaks in RDF plot, indicating that two hydrated shells have been formed. The peak intensity of $Li^+$ hydrated

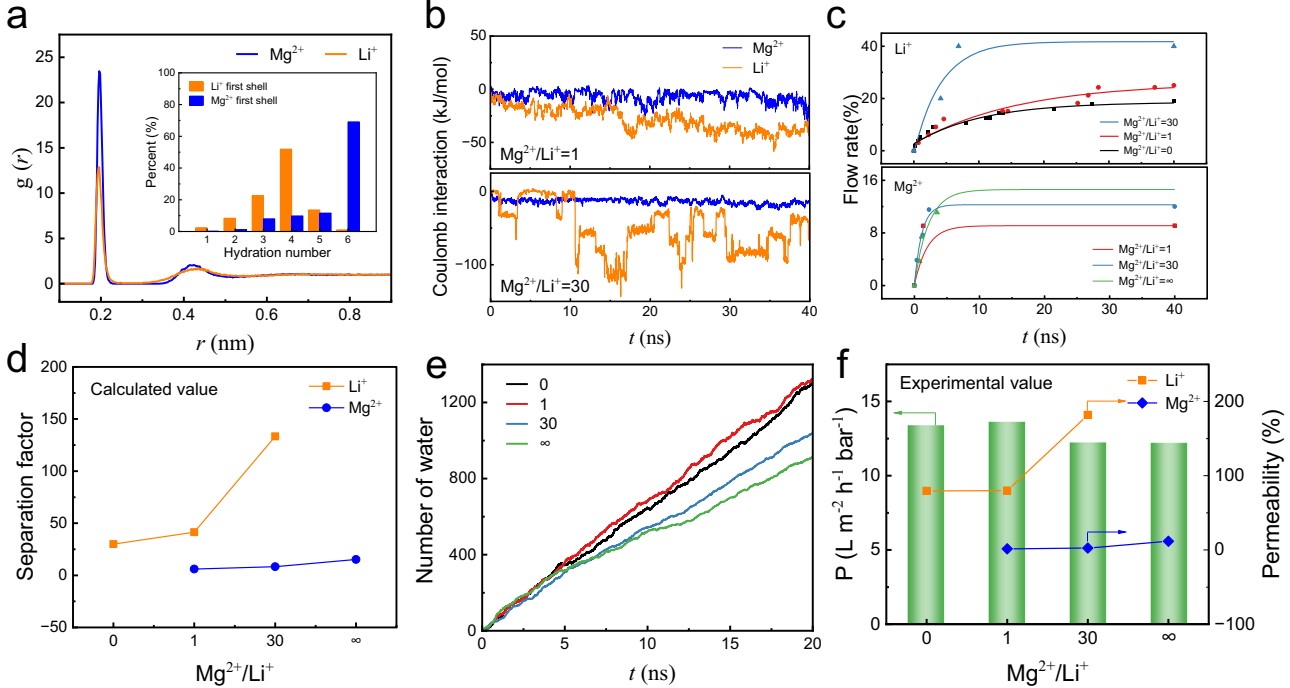

**Fig. 4 | Molecular simulation of Mg²⁺/Li⁺ separation. a** The Mg²⁺–oxygen and Li⁺–oxygen RDFs in water and the coordination number (Nc) distributions of the first hydration shells. **b** The average coulomb interaction between ions and AIP-PA membrane. **c** The flow rate of Li⁺ (upper) and Mg²⁺ (below), (**d**) the calculated separation factor (representing the ratio of the permeated concentration to the original concentration of specific ion.) of Mg²⁺ and Li⁺, and (**e**) the water flow in MD process. **f** permeation of Li⁺ and Mg²⁺ as well as water at different ratio of Mg²⁺/Li⁺ in experimental process (salt concentration: 2000 ppm; applied pressure: 4 bar).

shell is weaker, revealing a smaller hydration energy which is easier to dehydrate[42]. The hydration number distribution of Mg²⁺ and Li⁺ (the inset in Fig. 4a and Supplementary Fig. 48) also proves this point. In bulk solution, the first shell coordination numbers of Mg²⁺ and Li⁺ are concentrated in the range of 5–6 and 3–5 with average values of 5.39 and 3.68, respectively[43]. This result illustrates that Mg²⁺ has a considerable electrostatic interaction with water molecules, and the hydration group of Mg²⁺ is larger than that of Li⁺. In this case, more coordinated water molecules have to be stripped of when Mg²⁺ passes through the pore channels, which thus encounters a higher energy barrier than Li⁺ [44].

As shown in Fig. 4b, under different Mg²⁺/Li⁺ ratios, the coulomb interaction between the Li⁺ and AIP-PA membrane is higher than that of Mg²⁺, suggesting the more Li⁺ permeated into the membrane and interacted with the membrane. Moreover, the electrostatic interaction becomes stronger with the increase of Mg²⁺/Li⁺ ratios, resulting in an enhanced flow rate of Li⁺, whereas, the increase of Mg²⁺ transmittance is due to the initial concentration difference (Fig. 4c). It should be specially explained that the increasing range of Li⁺ permeability is much larger, leading to an excellent magnesium/lithium selectivity even at high Mg²⁺/Li⁺ ratio (Fig. 4d)[45]. In addition, Mg²⁺ blocking around the AIP-PA membrane pore was monitored at high Mg²⁺/Li⁺ ratio which reduces the water permeate flux (Fig. 4e). One interesting thing is that the simulated separation performance is in good agreement with the experimental results (Fig. 4f). Another simulation of transport behavior through a 6 nm-thick AIP-PA membrane displays similar results (Supplementary Figs. 49, 50). In a word, the synergy of hydrated shell differential, the difference in interaction between ions and membrane, and the refined pore structure endows the AIP-PA membrane with a sub-1 Å sieving property for excellent Mg²⁺/Li⁺ separation.

In conclusion, a unique solid-liquid anhydrous interfacial polymerization (AIP) was developed to reform the PA membrane-forming process. Due to the complete elimination of side reactions, and the intensive and ordered condensation reaction caused by the amine

molecules in compact arrangement, a selective PA layer with a narrow pore-size distribution capable of precise ionic sieving was constructed on the commercial porous substrate. The resulting AIP-PA membrane exhibited excellent separation selectivities of 78.3 and 29.2 for Mg²⁺/Li⁺ and Cl⁻/SO₄²⁻, respectively, benefiting from the synergy of hydrated shell differential, the difference in interaction between ions and membrane, and the refined pore structure. The water permeate flux is high up to 13.6 L m⁻² h⁻¹ bar⁻¹ and the overall performance is superior to that of the most PA membranes thus far in the literature. The AIP-PA membrane also displayed a long-time operational stability. Given by the sub-1 Å ionic sieving accuracy and flexibility of the AIP strategy, our work therefore provides an insight for the construction of high-precision PA membranes in various separation processes.

## Methods

### Materials and chemicals

Polyacrylonitrile ultrafiltration membrane (PAN50, molecular weight cut off -50000 g·mol⁻¹) was purchased from GUOCHU (Xiamen) Technology, China. Piperazine (PIP, ReagentPlus®, 99%) and trimesoyl chloride (TMC, AR, 98%,) obtained from Sigma-Aldrich, China. Anhydrous sodium sulfate (Na₂SO₄, AR, 99%) and sodium chloride (NaCl, AR, 99.5%), anhydrous calcium chloride (CaCl₂, AR, 99%), magnesium chloride anhydrous (MgCl₂, AR, 99%), barium chloride (BaCl₂, AR, 99%), nickel chlorideand (NiCl₂, AR, 99%) potassium chloride (KCl, AR, 99.5%) were provided by Sinopharm Chemical, China. anhydrous magnesium sulfate (MgSO₄, AR, 99%) was provided by XILONG Chemical, China. Deionized water (Conductivity <3 μS) was purchased from WAHAHA Group, China. Anhydrous ethanol (AR, 99.7%) was obtained from Tianjin Fuyu Fine Chemical, China; N-hexane (AR, 97%) and PEG1000 (AR, 99%) were provided by FUCHEN (Tianjin) Chemical, China. PEG200 (AR, 99%) was purchased from Shanghai Aladdin Biochemical Technology, China. Iodine (AR, 99.8%), PEG400, PEG600, Iodine (I₂), potassium iodide (KI) barium chloride (BaCl₂, AR, 99%), Cobalt chloride (CoCl₂, AR, 99%), Rubidium chloride (RbCl, 99%),

Caesium chloride (CsCl, 99%) and Gadolinium chloride (GdCl$_3$, AR, 99%) were obtained from Shanghai Macklin Biochemical Technology, China. Lanthanum chloride (LaCl$_3$, 99%), Samarium chloride (SmCl$_3$, 99%) was provided by Shanghai Bide Pharmaceutical Technology Co. China. Monolayer graphene oxide aqueous dispersion (2 mg mL$^{-1}$) was provided by Jiangsu Xianfeng Nanomaterials Technology Co. China. Copper mesh and sample stage are provided by Beijing KEHUA-JINGWEI Technology, China. Microscope slides were obtained from SAIL BRAND, China.

### Membrane preparation

**Pretreatment.** The PAN substrate was immersed in a 30% ethanol solution for 4 h, and then rinsed with deionized water for use.

**Conventional interfacial polymerization (CIP).** The pretreated substrate was immersed in a 1 wt% PIP solution for 30 s before being removed. The substrate was tilted around 60 °C to allow the aqueous solution on the surface to glide off. The filter paper was then used to clean and dry the non-sliding liquid and visible water droplets on the surface. The dry substrate was soaked in 0.2 wv% TMC solution for 30 s and then dipped in n-hexane solution for 30 s to remove the unreacted TMC. Finally, it was placed in a 60 °C oven for 3 min before being removed and stored in deionized water.

**Anhydrous interfacial polymerization (AIP).** The pretreated substrate was first dried with filter paper before being placed on a volatilization apparatus. Prior to that, a specific amount of piperazine was precisely weighed into the volatilization device. Then, the volatilization apparatus was immersed in a temperature-controlled water bath. After a period of volatilization, the substrate was taken out and immersed in a 0.2 wv% TMC n-hexane solution for 30 s to react. Following the reaction, it is soaked in an n-hexane solution to remove unreacted TMC, and then placed in an oven at 60 °C for 3 min before being immersed in deionized water for preservation.

### Characterizations

X-ray photoelectron spectroscopy (Thermo Scientific K-Alpha+, Thermo Fisher) was used to measure the element content of the active layer on the membrane surface. Total reflection Fourier transform infrared spectrometer (Bruker Tensor27) was used to scan the functional groups on the surface of the film. The scanning range was from 4000 cm$^{-1}$ to 600 cm$^{-1}$, and the data were normalized. Scanning electron microscope (JSM7610F) was used to photograph and analyze the surface and cross-section structure of the membrane, and the cross-section of the membrane is obtained by liquid nitrogen embrittlement. Before SEM observation, all samples must be sprayed with gold to enhance the conductivity. The hydrophilicity and hydrophobicity of the composite membrane surface were characterized by SDC-300 contact angle tester. The dried samples were cut into long strips and glued on the glass slide with double-sided adhesive. The water drop volume was 0.1 μL by the setting drop method. The contact angle of the membrane surface was measured five times for each sample, and the arithmetic mean value was taken. The three-dimensional morphology of the membrane surface was characterized by Bruker-Fastscan from Germany Brooke Company, and the surface roughness was obtained by NanoScope Analysis software. The surpass zeta potentiometer from the German Anto Paar Company was used to analyze the surface charge of the composite membrane. The Thermogravimetric Analysis from Japan Hitachi TG/DTA6300 was used to measure the volatile amount of amine monomer. UV-Vis spectrophotometer (UV Bluestar A) was used to measure the concentration of neutral substances. Conductivity meters (KEDIDA CT3030) were used to test the concentration of a single solution. Ion chromatography (Thermo Scientific ICS-900) was used to test the ionic concentration of the feed and permeate mixed salt solution. Positron annihilation doppler broadening spectroscopy from Institute of High Energy Physics Chinese Academy of Sciences was used to test polymer membrane pore size and free volume.

### Membrane performance evaluation

A Hangzhou Saifei Membrane Separation Technology Co., Ltd. assessment device (SF-SB) was used to test the performance of the nanofiltration membrane. The effective membrane area is approximately 7.1 cm². Three membrane tanks containing three different kinds of nanofiltration membranes were assembled and tested simultaneously. The salt ion concentration in the feed solution was 1 g L$^{-1}$, the applied pressure was set at 4 bar, and the cross-flow rate was set at 60 LPH. Prior to the nanofiltration experiment, the device was operated with deionized water for a while. Once the system was stable, the measurement started. The permeate liquid was collected and its mass change over time was used to estimate the permeation flux. The conductivity of the feed and permeate solutions was used to measure the ion selectivity. Three independent AIP procedures were conducted to prepare parallel membranes sample, and tests of each sample were repeated three times.

Equation (1) is used to calculate the rejection ($R$), where $C_f$ and $C_p$ indicate the conductivity of the feed and permeate solutions, respectively.

$$R = \left(1 - \frac{C_P}{C_f}\right) \times 100\% \tag{1}$$

The flow of $J$ is determined using formula (2), where $A$ is the effective membrane area, $\rho$ is the density of the permeate solution, and $w$ is the mass of the permeate solution at a given time $t$. Considering that the permeate solution is diluted, the density can be roughly estimated as 1 g mL$^{-1}$.

$$J = \frac{w}{\rho A t} \tag{2}$$

Permeance represents the flux per unit pressure, which is calculated by formula (3), where $\Delta P$ represents the pressure exerted on the nanofiltration membrane during nanofiltration evaluation.

$$Permeance = \frac{J}{\Delta P} \tag{3}$$

The ratio of the transmissions of two solutes through the membrane is defined as the selectivity ($S$). It can be calculated with the rejection difference between the two solutes.

$$S = \frac{1 - R_A}{1 - R_B} \tag{4}$$

where $R_A$ and $R_B$ represent the rejections of solute A and B, respectively.

### Determination of MWCO, pore size and pore-size distribution of membrane

A sequence of neutral organic molecules with increasing molecular weight can be used to evaluate the pore size of membrane. PEG-200, PEG-400, PEG-600, and PEG-1000 were utilized as neutral organic chemicals in this investigation to determine the pore size. Each organic solution had a concentration of 100 ppm, a cross-flow rate of 60 LPH, and an applied pressure of 4 bar. The concentration of PEG was measured using the UV-barium chloride method. Before testing, dilute the sample 10-20 times to make the absorbance between 0.2-0.8. Then add 1 ml of 0.05 M iodine standard solution and 1.2 ml of 5% barium chloride solution to 5 ml of sample solution. After 10 min of color

development, absorbance measurements were performed at a wavelength of 610 nm.

When the rejection is equal to 90%, the molecular weight is known as MWCO. The probability density function (PDF) used to calculate the average pore size distribution curve is based on the following premises: (1) There are no spatial or hydrodynamic interactions between these neutral organic substances and membrane pores; (2) The average pore size of the membrane is equal to the Stokes radius of organic solute with 50% rejection; and (3) The average pore size distribution of the membrane was calculated by the geometric standard deviation (p) of the PDF curve, which is defined as the ratio of the radius of solute molecule with rejection of 83.14% to the radius of solute molecule with the rejection of 50%[46–48].

$$\frac{dR(d_p)}{dr_p} = \frac{1}{r_p \ln \sigma_p \sqrt{2\pi}} \exp\left[-\frac{(\ln d_p - \ln \mu_p)^2}{2(\ln \sigma_p)^2}\right] \quad (5)$$

Where, $\mu_p$ represents the average pore size of the membrane, $\sigma_p$ represents the set standard deviation of PDF curve, $r_p$ represents the Stokes radius of neutral organic matter. The Stokes radius of these molecules is positively related to their molecular weight[46–48].

$$\log\left(r_p\right) = -1.4962 + 0.4654 \log(M_W) \quad (6)$$

Where, $M_W$ is the molecular weight of neutral organic compounds. The relationship between molecular radius and molecular weight of PEG is as follows[47,48].

$$r_s = 16.73 \times 10^{-12} \times M_w^{0.557} \quad (7)$$

## Molecular simulation

As shown in Supplementary Fig. 45, our simulation system contains two graphite sheets as pistons to apply external pressure, a PA porous membrane system, and two solution reservoirs. The dimensions of the simulation systems are about $5.54 \times 5.65 \times 28$ nm$^3$, and the negatively charged PA polymers randomly stack in the box forming a 4 nm thickness membrane with a few void size distributions. It should be emphasized that due to the homophaneous structure, to some extent, the 4 nm-thick atomic structure can represent the elementary mass transfer unit in the real membrane. The formed maximum pore size of PA membrane is about 0.81 nm. According to the experimental measurement (Supplementary Fig. 6), the polymer chain was negatively charged in this work to mimic the chemical environment inside the PA membrane[49]. The PA membrane was located at the center of the simulation box and connected to two chambers. The feed reservoir (left) was filled with mixed MgCl$_2$ and LiCl solution, and the concentration of the mixed solution was shown in Supplementary Table 8. It should be noted that the concentration used in the simulation is ten times higher than in the experimental to obtain more statistically significant data. In order to reduce the solid-liquid-gas contact interaction, the permeate reservoirs (right) also contains some pure water solution. Firstly, after the system was energy minimized, 1 ns equilibrium simulations were done to gain stable structure by applying a pressure of 1 bar on both sides of the piston to compress the polymer membrane. And the stable polymer membrane had a few void size distributions with a maximum van der Waals (vdW) pore size is about 8.1 Å. Finally, 40–60 ns non-equilibrium simulations were carried out under an external of 150 MPa. The external pressure between the two sides of membrane is generated by applying a constant force on the two sliding but otherwise rigid graphene[50]. The results were visualized using Visual Molecular Dynamics[51].

All MD simulations were performed by Gromacs2018.1[52], where TIP3P model is used for water. And all the particles were described by the CHARMM37 force field[53]. The electrostatic interactions were evaluated by the particle-mesh Ewald method[54] with a real-space cutoff of 14 Å. The vdW interactions and short-range repulsions between $i$ and $j$ atoms are modeled by Lennard-Jones (LJ) interactions with a cutoff of 14 Å, and evaluated by the Lorentz-Berthelot rules, $\varepsilon_{ij} = (\varepsilon_i \varepsilon_j)^{1/2}$ and $\sigma_{ij} = (\sigma_i \sigma_j)^{1/2}$, where $\varepsilon_{ij}$ are the effective well depths and $\sigma_{ij}$ are the minimum positions. Initially, the systems were energy minimized, thermalized at $T = 298$ K, and then non-equilibrium simulations were carried out in a constant number of particles, volume, and temperature (NVT) ensemble with periodic boundary conditions applied in $xyz$ direction. The temperature was controlled by V-rescale method with a relaxation time of 0.1 ps. In this work, the salt rejection of Mg and Li are defined by $(1\text{-}C_P/C_f) \times 100\%$, where $C_f$ and $C_p$ are the concentrations of ions in the feed and permeate reservoirs, respectively, where half of the water has flowed from the feed reservoir to the permeate reservoir[42]. To further verify the accuracy, another simulation regarding Mg$^{2+}$/Li$^+$ through a 6 nm-thick PA membrane was also performed and the procedure was similar to that of the 4 nm-thick membrane.

## Data availability

All data that support the findings of this study are available within the paper and its Supplementary Information or from the corresponding author upon request. Source data are provided with this paper.

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

## Acknowledgements

This work was financially supported by the National Key Research & Development Program of China (2022YFB3804802 (H.M and H.W.F)) and the National Natural Science Foundation of China (22108010 (H.W.F)).

## Author contributions

H.M. and H.W.F. conceived the concept and supervised the studies. G.J.Z. performed the membrane fabrication and characterization experiments. Z.Q. helped the performance experiments. H.Q.G. carried out the molecular dynamic simulations. All authors analyzed results and commented on the manuscript. G.J.Z., H.W.F. and H.M. wrote the paper.

## Competing interests

The authors declare no competing interests.
