## [Peer Review File · Nature Communications]

Anhydrous interfacial polymerization of sub-1 Å sieving
Polyamide membraneReviewers' Comments:

Reviewer #1:

Remarks to the Author:

The manuscript describes a quite creative approach for the preparation of thin film composite membranes. Instead of using the classical interfacial polymerization with aqueous and organic monomer solutions, one of the monomers was deposited by sublimation on a porous support and the reaction in contact with the organic solvent solution was then promoted. The results are encouraging, and I recommend the publication.

I just believe that Figure 1 does not clearly represent well the process. I do not see why a more ordered distribution of monomer would occur with the sublimation as apparently seen in Figure 1b. I believe it is still random but potentially with a higher density of distribution. While in a classical IP the most probable contact happens where the pores are filled with the aqueous phase, in the new method the amine-functionalized monomer will be rather distributed on the polymer matrix forming the porous support. The two methods are therefore in a way the negative of each other. If a gutter layer will be added between the porous support and the monomer layer, an even dense distribution of monomers and maybe tighter membrane would be form. At the same time, if the support would be more porous, at some point the membrane would not be formed. Could the authors please comment on that? Could they please try one experiment with a gutter layer?

Please check figure 4d. The x-axis might not be correct. It should not be permeability and not (%).

Reviewer #2:

Remarks to the Author:

1. The authors reported an anhydrous interfacial polymerization at the solid-liquid interface. The absence of water avoids the hydrolysis of acyl chloride during the IP process, resulting in a PA layer with more uniform pore size distribution than a conventional IP reaction performed at the water/oil interface. However, Liang et al. have realized a similar uniform pore structure of the PA NF membrane with the SARIP process. The data organization of this work is also identical to the work reported by Liang et al. Moreover, compared with the SARIP and conventional IP processes, the strategy proposed by the authors in this work is challenging to scale up. Therefore, I think that the importance of this work for the community of membrane technologies is low.
2. During the IP reaction between PIP and TMC, the PIP monomers are proposed to diffuse into the hexane and react with the TMC at the hexane phase near the interface. Due to the immiscibility between the water and hexane, there is little chance for the water to induce the hydrolysis of the acyl chloride group during the reaction. Because the trans-interface diffusion rate of PIP is much slower than the reaction rate, the IP reaction is far from the thermal-equilibrium state. The slow diffusion of PIP results in the molar ratio of PIP and TMC within the reaction far from stoichiometric equilibrium. As a result, the reaction products of PIP and TMC present heterogeneous microstructure and cause in turn the PA layer with a relatively wide pore size distribution.

Responses to Reviewer 1's Comments:

General comments: The manuscript describes a quite creative approach for the preparation of thin film composite membranes. Instead of using the classical interfacial polymerization with aqueous and organic monomer solutions, one of the monomers was deposited by sublimation on a porous support and the reaction in contact with the organic solvent solution was then promoted.

The results are encouraging, and I recommend the publication.

Response: We appreciate the reviewer's positive comments and high affirmation of our work. According to the reviewer's suggestions, we have supplemented more characterizations and performance data about the membrane prepared through a gutter layer. Structural analysis together with related discussions have been added into the revised manuscript. **Fig. 1b** has been appropriately modified, and the y-axis in **Fig. 4d** has been corrected. Please see the detailed point-by-point response as below:

Specific comments:

Comment 1: I just believe that Figure 1 does not clearly represent well the process. I do not see why a more ordered distribution of monomer would occur with the sublimation as apparently seen in Figure 1b. I believe it is still random but potentially with a higher density of distribution. While in a classical IP the most probable contact happens where the pores are filled with the aqueous phase, in the new method the amine-functionalized monomer will be rather distributed on the polymer matrix forming the porous support. The two methods are therefore in a way the negative of each other. If a gutter layer will be added between the porous support and the monomer layer, an even dense distribution of monomers and maybe tighter membrane would be form. At the same time, if the support would be more porous, at some point the membrane would not be formed. Could the authors please comment on that? Could they please try one experiment with a gutter layer?

Response 1: Many thanks for the reviewer's comments and valuable suggestions. Indeed, as the reviewer mentioned that the deposited monomer layer with the sublimation is random but potentially with a higher density of distribution. Therefore, we appropriately modified the schematic elements in **Fig. 1b** (please see it as follows) to make the representation more accurate.

Figure 1. Schematic illustration of CIP and AIP processes. Scheme depicting the preparation of (a) CIP-PA and (b) AIP-PA membranes.

Furthermore, we also believe that adding gutter layer could be a valid way to obtain a denser distribution of monomers, leading to a tighter membrane. According to the reviewer's suggestions, new membranes have been prepared by constructing a GO layer as the gutter layer on the substrate, and followed by the same procedure as other membranes. As a control, the membrane performance was also examined by rejecting different salt solutions. Please see the detailed experimental processes and results as follows:

Figures together with relative discussions added into the Supplementary Information:

Supplementary Figure 28. A schematic illustration showing the preparation process of GL-PA membrane.

Preparation of GO gutter layer: Firstly, 1 ml of 2 mg/mL graphene oxide (GO) aqueous solution was accurately prepared, then 1999 mL of deionized water was added. The solution was therefore diluted to be 1 ppm, and then subjected to stirring and sonification for 1 h each. Finally, 20 mL of GO dispersion was taken and filtered onto the surface of PAN substrate by the assistance of vacuum filtration to prepare the GO gutter layer.

Supplementary Figure 29. Scheme depicting the preparation of GL-PA membrane.

Preparation of GL-PA membrane: The substrate with GO gutter layer was first dried with filter paper before being placed on a volatilization apparatus. Prior to that, a specific amount of piperazine was added into the volatilization device. Thereafter, the volatilization apparatus was immersed in a temperature-controlled water bath for a period of volatilization. Afterwards, the substrate was taken out and immersed in a 0.2 wv% TMC *n*-hexane solution for 30 s to finish the interfacial polymerization. Finally, the membrane was soaked in an *n*-hexane solution to remove unreacted TMC, and then placed in an oven at 60 °C for 3 minutes before test.

Supplementary Figure 30. Digital Photo of the prepared GL-PA membrane.

Supplementary Figure 31. ATR-FTIR survey of PAN substrate and the prepared GL-PA membranes.

As shown in **Supplementary Fig. 31**, the amide group formed from the reaction between solid PIP and TMC in *n*-hexane is responsible for the new stretching vibration peak of C=O bond at 1630 cm^{-1} and of C-N bond at 1416 cm^{-1} , confirming the successful synthesis of the GL-PA layer by AIP.

Supplementary Figure 32. SEM image of GO deposited on the substrate.

Supplementary Figure 33. SEM image of GL-PA membrane.

Supplementary Figure 34. AFM image of GL-PA membrane.

As shown in **Supplementary Fig. 32**, compared to the porous PAN substrate, the construction of a GO layer on the PAN substrate resulted in a smoother surface. This gutter layer makes the distribution of volatile PIP molecules more uniform and compact. From **Supplementary Fig. 33 and 34**, the gutter layer could avoid the intrusion of PA layer into the pores, and allows PA to grow on the surface of the gutter layer, enhancing the denseness of the polyamide membrane.

Supplementary Figure 35. MWCO and pore-size distribution measured by rejecting PEG with different molecular weights.

The MWCO of the prepared GL-PA membrane is evaluated as 300 Da and the average pore radius is 2.31 Å. Both are smaller than 340 Da and 2.53 Å of AIP-PA membrane. These results prove that the PA membrane prepared with the gutter layer is denser and possibly has a stronger size sieving effect.

Supplementary Figure 36. Salt reject performance of GL-PA membrane.

As shown in **Supplementary Fig. 36**, the rejection of GL-PA membrane for two sulfates were 97.92% (Na₂SO₄) and 98.60% (MgSO₄), respectively. The retention effect for divalent cationic chloride salts were all above 94%, especially 94.83% for BaCl₂, which was slightly higher than 93.59% for the AIP-PA membrane. The rejection of GL-PA membrane for LiCl and NaCl were 35.06% and 30.89%, respectively, which were at least 9% higher than those of AIP-PA membrane. These results imply that the introduction of the interlayer enabled a denser structure with narrower pores inside the resulting PA membrane, which was beneficial to the rejection of small molecules.

In addition, we also tried to prepare the membrane on a more porous substrate through the AIP strategy. For this purpose, we used a polyethersulfone (PES) membrane with a wider pore size of 0.22 μm as the substrate, the preparation process is the same as that of other AIP membranes. As can be seen from AFM images (**Supplementary Fig. 37**), the PA layer can cover the non-porous area of the substrate, but can not completely cover the macropores, which also implies the possible assistance of gutter layer for our AIP strategy.

Figures together with relative discussions added into the Supplementary Information:

Supplementary Figure 37. (a) 2D and (b) 3D AFM images of PA membrane prepared on PES microfiltration substrate. (the pore size of PES substrate: 0.22 μm)

Based on the reviewer's comments, we have added the follow content into the revised manuscript:

“Additionally, introducing a gutter layer is an emerging approach to effectively tune the structure and properties of PA membrane by improving amine distribution at the interface and avoiding pore infiltration of amine.^{11,16} We also incorporated a graphene oxide layer as the gutter layer to further regulate the AIP process (**Supplementary Fig. 28 & 29**) in an attempt to obtain more compact and more refined pore structure in the resulting membrane (GL-PA membrane) (**Supplementary Fig. 30-34**), by making the distribution of sublimated PIP molecules on porous substrate more uniform and denser. As expected, **Supplementary Fig. 35** displays a smaller MWCO of 300 Da and a smaller pore size (about 4.62 Å) than that of the AIP-PA membrane (**Fig. 2j**). Based on this, the GL-PA membrane showed higher salt rejections, despite a little decrease in water permeance flux (**Supplementary Fig. 36**). For example, the rejection for Na_2SO_4 and MgSO_4 can reach up to 97.92% and 98.60%, respectively, and the rejections for divalent cationic chloride salts were all above 94%. Moreover, the GL-PA membrane exhibited rejections of 35.06% and 30.89% for LiCl and NaCl, respectively, which were at least 9% higher than those of AIP-PA membrane. These results imply that the gutter layer enabled a tighter structure with narrower pores in the resulting GL-PA membrane, which was beneficial for selective rejection of solutes. A negative result of preparation of PA layer on a more porous substrate of polyether sulfone (with a wide pore size of 0.22 μm) (**Supplementary Fig. 37**) also suggests the possible assistance of gutter layer to the AIP strategy.”

Comment 2: Please check figure 4d. The x-axis might not be correct. It should not be permeability and not (%).

Response 2: Thanks for reviewer's kind correction. The permeability (%) on the y-axis of Fig. 4d has been replaced with permeation (%), and the permeation represents the ratio of the permeated concentration to the original concentration of specific ion. Please see the revised Fig. 4d as follow:

Figure 4. Molecular simulation of Mg²⁺/Li⁺ separation. (a) The Mg²⁺-oxygen and Li⁺-oxygen RDFs in water and the coordination number (Nc) distributions of the first hydration shells. (b) The average coulomb interaction between ions and AIP-PA membrane. (c) The flow rate of Li⁺ (upper) and Mg²⁺ (below), (d) the calculated permeation of Mg²⁺ and Li⁺, and (e) the water flow in MD process. (f) Permeation of Li⁺ and Mg²⁺ as well as water at different ratio of Mg²⁺/Li⁺ in experimental process (salt concentration: 2000 ppm; applied pressure: 4 bar).

Responses to Reviewer 2's Comments:

Comment 1: The authors reported an anhydrous interfacial polymerization at the solid-liquid interface. The absence of water avoids the hydrolysis of acyl chloride during the IP process, resulting in a PA layer with more uniform pore size distribution than a conventional IP reaction performed at the water/oil interface. However, Liang et al. have realized a similar uniform pore structure of the PA NF membrane with the SARIP process. The data organization of this work is also identical to the work reported by Liang et al. Moreover, compared with the SARIP and conventional IP processes, the strategy proposed by the authors in this work is challenging to scale up. Therefore, I think that the importance of this work for the community of membrane technologies is low.

Response 1: Many thanks for the reviewer's comments on this manuscript. As the reviewer mentioned, Liang and co-workers did an excellent work on the construction of PA NF membrane with highly uniform pore structure (*Nat. Commun.* **11**, 2015 (2020)), which also has already been cited in our manuscript (Ref. 13). However, there are significant differences between this study and our work, including design concept, strategy and technological means, despite both aim at the challenge of realizing sub-1 Å precision separation. The main differences are listed as below:

1) The specific preparation method is different. Liang et al. obtained PA active layers with sub-1 Å pores by adding surfactants to the aqueous phase based on the CIP to promote the diffusion of monomers. In our work, we developed a different strategy of anhydrous solid-liquid IP to prepare narrow pore size PA NF membranes by removing the aqueous phase.

2) The forming mechanism of membrane structure is different. For the work by Liang et al, the dynamic self-assembling network of surfactants facilitates faster and more uniform diffusion of amine monomers at the water/hexane interface during IP, resulting in the formation of PA active layers with more uniform sub-nanometer pores than CIP. In our work, due to the absence of the aqueous phase, the anhydrous solid-liquid IP, on the one hand, avoids the hydrolysis of the acyl chloride to produce non-cross-linkable sites and structural defects in the polyamide; on the other hand, the diffusion step of the amine monomer in the aqueous phase is omitted, and the volatile solid amine molecules

directly undergo a rapid and violent Schotten-Bauman reaction with the acyl chloride monomer at the interface, resulting in a narrow pore size PA film.

3) The research focus is different. Liang et al. put more emphasis on the mechanism of surfactant action in the IP process, *i.e.*, the enrichment of PIP molecules by sulfonic acid groups in SDS and the regularization of PIP monomer diffusion by SDS-ordered monomolecular membranes. Whereas, in our work, in addition to the membrane forming mechanism, we also paid much attention on the magnesium/lithium separation mechanism by using the sub-1 Å sieving PA membranes. The synergy of hydrated shell differential, the difference in interaction between ions and membrane, and the refined pore structure endow the AIP-PA membrane with a sub-1 Å sieving property for excellent $\text{Mg}^{2+}/\text{Li}^{+}$ separation.

In a word, liang et al. proposed a simple and effective strategy to promote the interphase diffusion of amine monomers to obtain PA films with sub-1Å sieving effect. Nevertheless, we proposed a different strategy to enable the IP reaction to be more intensive and ordered to obtain dense PA nanofiltration membranes. In this case, we considered to eliminate the aqueous phase and omit the diffusion step of amine monomer to react directly with acyl chloride, and thereby explored an AIP approach. Given the uniqueness and the result of excellent separation performance, the AIP strategy is expected to provide inspiration for engineering advanced PA membranes and other membrane-based functional materials.

In addition, we also demonstrated the scaling-up ability of our AIP strategy by preparing larger-area membranes with an area of 314 cm² through an improved equipment (as shown in **Supplementary Fig. 38-41**). Such membranes were placed inside a custom-made membrane module for NF measurement (**Supplementary Fig. 42 and 43**). The results show that the membrane retained up to 95.88% of MgSO_4 with a treatment capacity of 601.39 L m⁻² d⁻¹, and the difference between the retention of Mg^{2+} and Li^{+} by the membrane was still above 70%, which can effectively separate Mg^{2+} and Li^{+} , demonstrating the great potential in industrial application (**Supplementary Fig. 44**).

Based on the reviewer's comments, we have added related discussions into the revised manuscript (please also see it as below), and **Supplementary Fig. 38-44** have been added into the Supplementary Information. We would appreciate very much if the

reviewer could understand our further explanations and efforts.

Contents added into the revised manuscript:

“Further, we demonstrated the scaling-up ability of AIP strategy by preparing larger-area membranes with an area of 314 cm² using an improved equipment (**Supplementary Fig. 38-41**). Such membranes were placed inside a custom-made membrane module for NF measurement (**Supplementary Fig. 42 and 43**). This membrane module could retain MgSO₄ up to 95.88% together with a treatment capacity of 601.39 L m⁻² d⁻¹. Meanwhile, effective separation of Mg²⁺ and Li⁺ could also be realized due to the large difference of rejection (above 70%) for Mg²⁺ and Li⁺, demonstrating the potential of the AIP-PA membranes in industrial application (**Supplementary Fig. 44**).”

Figures added into the Supplementary Information:

Supplementary Figure 38. Large-area AIP-PA membrane preparation device.

Supplementary Figure 39. Photo of large-area AIP-PA membrane.

Supplementary Figure 40. SEM images of the large-area AIP-PA membrane.

Supplementary Figure 41. Water contact angle images of the large-area AIP-PA membrane.

Supplementary Figure 42. Photo of large area membrane modules.

Supplementary Figure 43. Photo of the membrane module evaluation system.

Supplementary Figure 44. Salt rejection performance of large-area AIP-PA membrane. (operate pressure: 4 bar; salt concentration: 1g/L)

Comment 2: During the IP reaction between PIP and TMC, the PIP monomers are proposed to diffuse into the hexane and react with the TMC at the hexane phase near the interface. Due to the immiscibility between the water and hexane, there is little chance for the water to induce the hydrolysis of the acyl chloride group during the reaction. Because the trans-interface diffusion rate of PIP is much slower than the reaction rate, the IP reaction is far from the thermal-equilibrium state. The slow

diffusion of PIP results in the molar ratio of PIP and TMC within the reaction far from stoichiometric equilibrium. As a result, the reaction products of PIP and TMC present heterogeneous microstructure and cause in turn the PA layer with a relatively wide pore size distribution.

Response 2: Thanks for the reviewer's comments. IP containing the processes of reaction and diffusion is often an interesting research topic worthy to be discussed. First, for CIP, we believe that the acyl chloride monomers would inevitably contact water molecules at the immiscible two-phase interface, and therefore lead to the hydrolysis of acyl chloride group. This phenomenon and related statement can also be found in previous reported work.^{1,2} Actually, most of the PA membranes having strongly negative charges suggests the hydrolysis of acyl chloride during IP. In this case, hydrolysis of acyl chloride could also be one of the influence factors causing the PA layer with a relatively wide pore size distribution.

Second, the reviewer pointed out another main reason for the formation of PA layer with a relatively wide pore size distribution, "Because the trans-interface diffusion rate of PIP is much slower than the reaction rate, the IP reaction is far from the thermal-equilibrium state. The slow diffusion of PIP results in the molar ratio of PIP and TMC within the reaction far from stoichiometric equilibrium." We totally agree the reviewer's viewpoint and also have already mentioned it in our manuscript.

Overall, pore defects and wider pore distribution in the PA membrane are caused by at least two reasons analyzed above. Our AIP strategy can effectively avoid the problem of pore defects from hydrolysis of TMC in the presence of water molecules. More importantly, this process does not need the diffusion step of the amine monomer in the aqueous phase, endowing a direct reaction with the TMC in hexane at the solid-liquid interface, thus promoting the IP reaction to obtain PA membranes with a narrow pore size distribution. Actually, the related statements have been presented in our manuscript. Please see it as follows

"Fig. 1b and Supplementary Fig. 2 show the AIP process which is performed in such an expected way, including two steps. Firstly, PIP molecules were volatilized and adsorbed onto the surface of a polyacrylonitrile (PAN) ultrafiltration substrate to form a solid-phase layer. The PIP content at the interface could be controlled by altering the

volatilization temperature and time. Secondly, the AIP was carried out by immersing the PIP-containing PAN substrate into the TMC *n*-hexane solution. This anhydrous interface can entirely avoid the hydrolysis of TMC and the subsequent side reaction. Moreover, it could be reasonably speculated that the amine monomer molecules with no diffusion in aqueous phase directly reacted with the TMC in the *n*-hexane at the solid-liquid interface, which facilitated the IP reaction in an intensive and ordered manner”.

Reference:

1. Liu, C. *et al.* Interfacial polymerization at the alkane/ionic liquid interface. *Angew. Chem. Int. Ed.* **60**, 14636–14643 (2021).
2. Palling, D. & Jencks, W. P. Nucleophilic reactivity toward acetyl chloride in water. *J. Am. Chem. Soc.* **106**, 4869–4876 (1984).

Reviewers' Comments:

Reviewer #1:

Remarks to the Author:

I am satisfied with the revised version, except for Figure 4d.

Naming the y-axis "permeation" is not correct and must be changed.

The closest that we have to what is presented is "separation factor", although this might not be the exact definition commonly used in the literature. I suggest changing "permeation" to "separation factor" and clearly defining it in the text.

Reviewer #2:

Remarks to the Author:

The authors have attempted to address the reviewers' comments and enhance the manuscript; however, there are still notable concerns that require the authors' attention.

1. I have some reservations regarding the scalability of the AIP process. Despite attempts to create a larger membrane of 314 cm², it remains challenging to achieve continuous production through vapor-phase deposition of PIP, similar to the conventional IP process.
2. Another concerning issue is how to restrict the adsorption of volatilized PIP only at the surface of the PAN membrane rather than entering the pore. How to control the polymerization occurs only at the surface. This is very important to ensure the formation of thin-film composite structure. In theory, the pore of the PAN ultrafiltration membrane is much larger than the molecular size of PIP. It cannot prevent the PIP from entering the pore.
3. I noticed the MWCO of the resulting PA membrane obtained using AIP is around 340 Da, which is higher than many other PA membranes obtained using CIP reported in the literature (J. Membr. Sci. 2021, 627, 119142; J. Membr. Sci. 2023, 667, 121187; J. Membr. Sci. 2022, 643, 120056). This MWCO is similar to the commercial NF270 membrane. For the preparation of PA membrane using CIP in this work, the PIP concentration is 1 wt%, and PAN is used as the substrate. I believe that the pore size distribution of the PA membrane could be sharpened if the authors optimize the mass ratio of PIP and TMC. Therefore, this data cannot support the conclusion that the AIP can narrow the pore size distribution of the PA layer.
4. More importantly, many PIP-based PA membranes with smaller MWCO cannot realize the high rejection of MgCl₂. For instance, as reported in J. Membr. Sci. 2021, 627, 119142, the PA membrane with MWCO of 232 Da can only produce an MgCl₂ rejection of 60%. Further characterizations are needed to clarify the chemical and pore structure of the PA membranes.
5. The author uses MD to study the Li⁺ and Mg²⁺ transport behavior. The dimensions of the simulation systems are about 5.54×5.65×28 nm³ and the thickness of the PA membrane is only 4 nm in the simulated system. I think the size is too small to reflect the real membrane structure. At the very least, the authors should model the PA layer based on its thickness and pore size distribution. There is no any data to tell us the pore structure of the simulated PA membrane.
6. The MD study does not concern the counter ion, Cl⁻. During the actual separation process, the Cl⁻ transport significantly affects the ion transport of Li⁺ and Mg²⁺, which cannot be ignored. Without the study of Cl⁻ transport, the simulated system will lead to incorrect results and conclusions.
7. The process of transporting Li⁺ and Mg²⁺ through the PA membrane is intricate, and it depends on various factors like the pore structure, surface charge, and the chemical environment of the inner pore wall. The PA membrane made from AIP has a relatively large MWCO, and its pore size distribution is not narrower than other NF membranes. This PA NF membrane cannot reject MgCl₂ beyond 90%, as far as I know. The MD simulation alone cannot fully explain why this happens, and further investigation of the chemical environment inside the PA membrane is necessary.

Responses to Reviewer 1's Comments:

Comment 1: I am satisfied with the revised version, except for Figure 4d. Naming the y-axis "permeation" is not correct and must be changed. The closest that we have to what is presented is "separation factor", although this might not be the exact definition commonly used in the literature. I suggest changing "permeation" to "separation factor" and clearly defining it in the text.

Response 1: Thanks for reviewer's kind correction. Based on the reviewer's suggestion, the "permeation" on the y-axis of **Fig. 4d** has been replaced with "separation factor". The separation factor represents the ratio of the permeated concentration to the original concentration of specific ion. Please see the revised **Fig. 4d** and corresponding definition of "separation factor" in Figure caption as below:

Figure 4. Molecular simulation of Mg²⁺/Li⁺ separation. (a) The Mg²⁺-oxygen and Li⁺-oxygen RDFs in water and the coordination number (N_c) distributions of the first hydration shells. (b) The average coulomb interaction between ions and AIP-PA membrane. (c) The flow rate of Li⁺ (upper) and Mg²⁺ (below), (d) the calculated separation factor (representing the ratio of the permeated concentration to the original concentration of specific ion.) of Mg²⁺ and Li⁺, and (e) the water flow in MD process. (f) permeation of Li⁺ and Mg²⁺ as well as water at different ratio of Mg²⁺/Li⁺ in experimental process (salt concentration: 2000 ppm; applied pressure: 4 bar).

Responses to Reviewer 2's Comments:

The authors have attempted to address the reviewers' comments and enhance the manuscript; however, there are still notable concerns that require the authors' attention.

Comment 1: I have some reservations regarding the scalability of the AIP process. Despite attempts to create a larger membrane of 314 cm², it remains challenging to achieve continuous production through vapor-phase deposition of PIP, similar to the conventional IP process.

Response 1: Thank you for your valuable comments. Admittedly, the current continuous production of PA films by PIP volatilization is challenging compared to the conventional IP process. Over past two more months, we have been trying to resolve this problem and also consulted relative membrane-producing companies. There is a potential of the scalability by improving the AIP process and a preliminary scheme regarding the continuous production has been determined. The details will be presented in our future work.

Comment 2: Another concerning issue is how to restrict the adsorption of volatilized PIP only at the surface of the PAN membrane rather than entering the pore. How to control the polymerization occurs only at the surface. This is very important to ensure the formation of thin-film composite structure. In theory, the pore of the PAN ultrafiltration membrane is much larger than the molecular size of PIP. It cannot prevent the PIP from entering the pore.

Response 2: This is a good comment. We agree with that PIP will inevitably invade into the pore channel. However, on the one hand, since the saturation vapor pressure of PIP molecules is low, the volatilized PIP content is small and can not fully fill the pore channel. On the other hand, the amine monomers deposited on the surface of the PAN membrane is in compact arrangement which enabled the condensation reaction to be more intensive and fast, and the formed layer would prohibit the subsequent PIP invasion. Therefore, the invaded PIP will not only not impede the component permeation too much but endow the PA membrane a better binding strength with the PAN substrate. The structural characterizations such as Figure 2 and nanofiltration

performance could support this claim.

In addition, in order to prevent the PIP molecules from invading into the pore channel, we constructed a GO gutter layer to investigate the membrane structure and nanofiltration performance in depth. Actually, we have already supplemented this point in the previous submission, and the reviewer was satisfied with our revision. Please see again the specific experimental results as shown below, and the added discussion are marked in yellow in the revised manuscript.

Supplementary Figure 29. A schematic illustration showing the preparation process of GL-PA membrane.

Preparation of GO gutter layer: Firstly, 1 ml of 2 mg/mL graphene oxide (GO) aqueous solution was accurately prepared, then 1999 mL of deionized water was added. The solution was therefore diluted to be 1 ppm, and then subjected to stirring and sonification for 1 h each. Finally, 20 mL of GO dispersion was taken and filtered onto the surface of PAN substrate by the assistance of vacuum filtration to prepare the GO gutter layer.

Preparation of GL-PA membrane: The substrate with GO gutter layer was first dried with filter paper before being placed on a volatilization apparatus. Prior to that, a specific amount of piperazine was added into the volatilization device. Thereafter, the volatilization apparatus was immersed in a temperature-controlled water bath for a period of volatilization. Afterwards, the substrate was taken out and immersed in a 0.2 wv% TMC *n*-hexane solution for 30 s to finish the interfacial polymerization. Finally, the membrane was soaked in an *n*-hexane solution to remove unreacted TMC, and then placed in an oven at 60 °C for 3 minutes before test.

Supplementary Figure 30. Scheme depicting the preparation of GL-PA membrane.

Supplementary Figure 31. Photo of GL-PA membrane.

Supplementary Figure 32. ATR-FTIR survey of PAN substrate and GL-PA membranes.

As shown in **Supplementary Fig. 32**, the amide group formed from the reaction between solid PIP and TMC in *n*-hexane is responsible for the new stretching vibration peak of C=O bond at 1630 cm^{-1} and of C-N bond at 1416 cm^{-1} , confirming the successful synthesis of the GL-PA layer by AIP.

Supplementary Figure 33. SEM image of GO deposited on the substrate.

Supplementary Figure 34. SEM image of GL-PA membrane.

Supplementary Figure 35. AFM image of GL-PA membrane.

As shown in Supplementary Fig. 2, compared to the porous PAN substrate (Supplementary Fig.10a), the construction of a GO layer on the PAN substrate resulted in a smoother surface. This gutter layer makes the distribution of volatile PIP molecules more uniform and compact. From Supplementary Fig. 34 and 35, the gutter layer` reduces could avoid the intrusion of PA layer into the pores, and allows PA to grow on the surface of the gutter layer, enhancing the denseness of the polyamide membrane.

Supplementary Figure 36. MWCO and pore-size distribution measured by rejecting PEG with different molecular weights.

The MWCO of the prepared GL-PA membrane is evaluated as 300 Da and the average pore radius is 2.31 Å. Both are smaller than 340 Da and 2.53 Å of AIP-PA membrane. These results prove that the PA membrane prepared with the gutter layer is denser and possibly has a stronger size sieving effect.

Supplementary Figure 37. Salt reject performance of GL-PA membrane.

As shown in Supplementary Fig. 37, the rejection of GL-PA membrane for two sulfates were 97.92% (Na_2SO_4) and 98.60% (MgSO_4), respectively. The retention effect for divalent cationic chloride salts were all above 94%, especially 94.83% for BaCl_2 , which was slightly higher than

93.59% for the AIP-PA membrane. The rejection of GL-PA membrane for LiCl and NaCl were 35.06% and 30.89%, respectively, which were at least 9% higher than those of AIP-PA membrane. These results imply that the introduction of the interlayer enabled a denser structure with narrower pores inside the resulting PA membrane structure, which was beneficial to the rejection of small molecules.

Comment 3: I noticed the MWCO of the resulting PA membrane obtained using AIP is around 340 Da, which is higher than many other PA membranes obtained using CIP reported in the literature (J. Membr. Sci. 2021, 627, 119142; J. Membr. Sci. 2023, 667, 121187; J. Membr. Sci. 2022, 643, 120056). This MWCO is similar to the commercial NF270 membrane. For the preparation of PA membrane using CIP in this work, the PIP concentration is 1 wt%, and PAN is used as the substrate. I believe that the pore size distribution of the PA membrane could be sharpened if the authors optimize the mass ratio of PIP and TMC. Therefore, this data cannot support the conclusion that the AIP can narrow the pore size distribution of the PA layer.

Response 3: Thanks for your comments. As we all know that the MWCO of the membrane is not only related to the pore size distribution of the membrane itself, but also related to the cutoff substance used. At present, two series of substances are commonly used as MWCO calibration substances. One is glycerol, glucose, sucrose, and raffinose and other small molecules with more spherical shape. The other is polyethylene glycol with different molecular weights, which is more chain-like, so the MWCO of the two different series of substances will deviate from each other, i.e., and the calculated MWCO is usually larger with the latter. Moreover, since the pore size of the nanofiltration membrane is not completely uniform, and there is a certain pore size distribution. Therefore, purely relying on the MWCO does not reflect the real retention effect of the membrane, and the separation performance of the membrane will be proved by further fitting the average pore size distribution curve of the membrane. The specific method is described below:

When the rejection is equal to 90%, the molecular weight is known as MWCO. The probability density function (PDF) used to calculate the average pore size distribution curve is based on the following premises: (1) There are no spatial or hydrodynamic interactions between these neutral organic substances and membrane pores; (2) The average pore size of membrane is equal to the Stokes radius of organic solute with 50% rejection; and (3) The average pore size distribution of the membrane

was calculated by the geometric standard deviation (p) of the PDF curve, which is defined as the ratio of the radius of solute molecule with rejection of 83.14% to the radius of solute molecule with rejection of 50%.¹⁻⁴

$$\frac{dR(d_p)}{dr_p} = \frac{1}{r_p \ln \sigma_p \sqrt{2\pi}} \exp \left[-\frac{(\ln d_p - \ln \mu_p)^2}{2(\ln \sigma_p)^2} \right] \quad (1)$$

Where, μ_p represents the average pore size of the membrane, σ_p represents the set standard deviation of PDF curve, r_p represents the Stokes radius of neutral organic matter. The Stokes radius of these molecules is positively related to their molecular weight¹⁻³.

$$\log(r_p) = -1.4962 + 0.4654 \log(M_w) \quad (2)$$

Where, M_w is the molecular weight of neutral organic compounds. The relationship between molecular radius and molecular weight of PEG is as follows²⁻⁴.

$$r_s = 16.73 \times 10^{-12} \times M_w^{0.557} \quad (3)$$

The measured MWCO will be different because of the different equations corresponding to the Stokes radius for the two. For example, assuming that the nanofiltration membrane retains 90% of spherical molecules with a radius of 0.4 nm, the MWCO measured using glycerol series according to Eq. 2 would be 229 Da, while the MWCO measured using PEG series according to Eq. 3 would be 298 Da. For the three literature mentioned, we will analyze them one by one:

J. Membr. Sci. 2021, 627, 119142: In the article, the MWCO test was carried out using neutral substances such as glycerol, diethylene glycol and PEG, and the MWCO of the membranes prepared using CIP was measured to be 232 Da, with a pore diameter of 0.423 nm, which were indeed smaller than those of our AIP membrane. However, we are confused that the average pore radius of this membrane is 0.2115 nm, which is far smaller than the Stokes radius of Mg^{2+} (0.347 nm) and the hydration radius (0.423 nm), but the retention effect of this membrane on $MgCl_2$ is only 60%. Meanwhile, the literature (Nat. Commun. 2020, 11, 2015) has a pore radius of 0.28 nm, and its retention effect on $MgCl_2$ is as high as 95%.

J. Membr. Sci. 2023, 667, 121187: In the article, MWCO tests using PEGs of different molecular weights were performed. The CIP membrane (TFC0) in this paper has a MWCO of 505 Da (Figure S5), which is larger than our CIP and AIP membrane. The optional membrane (TFC0.6) in this paper has a MWCO of 810 Da (Figure S5) and a

rejection rate of 55.63% for MgCl₂ (Figure 5D). In contrast, the AIP-PA membrane in our paper had a MWCO around 340 Da, and its rejection of MgCl₂ was 93.9%. It is consistent with its trend.

J. Membr. Sci. 2022, 643, 120056: In this article, MWCO tests using glucose, sucrose, raffinose, PEG600, PEG1000 were performed. The glucose, sucrose, and neon oligosaccharides used are small spherical molecules, and the measured retention molecular weight is lower compared to that of PEG. In addition, the average pore radius of its TFC-0 was 0.248 nm, which is similar to that of our AIP-PA membrane with a pore radius of 0.25 nm. However, the retention rate of TFC-0 for MgCl₂ was only about 32% (Figure 9c), indicating that the wide pore size distribution of the membrane prepared by CIP would lead to Mg²⁺ permeation in spite of the smaller average pore size, whereas the membrane prepared by AIP could retain up to 93% of MgCl₂ due to its narrower pore size distribution.

In addition, we have already carried out CIP conditions before the comparison experiment, we fixed the TMC concentration as 0.2%, and by adjusting the PIP concentration and then adjusting the mass ratio of PIP and TMC, the following results can be obtained. From **Figure R1**, with the increase of PIP concentration, the retention effect of polyamide on sodium sulfate firstly increases and then tends to level off, while its water flux gradually decreases, taking into account the permeability and retention effect, when the PIP concentration of 1% is optimal.

Figure R1 Effect of PIP concentration on nanofiltration performance of polyamide membranes

Reference

1. Zhang, S., Fu, F. & Chung, T.-S. Substrate modifications and alcohol treatment on thin film composite membranes for osmotic power. *Chem. Eng. Sci.* **87**, 40–50 (2013).
2. Chen, G.-E. *et al.* Fabrication and characterization of a novel nanofiltration membrane by the interfacial polymerization of 1,4-diaminocyclohexane (DCH) and trimesoyl chloride (TMC). *RSC Adv.* **5**, 40742–40752 (2015).
3. Liang, Y. *et al.* Polyamide nanofiltration membrane with highly uniform sub-nanometre pores for sub-1 Å precision separation. *Nat. Commun.* **11**, 2015 (2020).
4. Wang, K. The effects of flow angle and shear rate within the spinneret on the separation performance of poly(ethersulfone) (PES) ultrafiltration hollow fiber membranes. *J. Membr. Sci.* **240**, 67–79 (2004).

Comment 4: More importantly, many PIP-based PA membranes with smaller MWCO cannot realize the high rejection of MgCl₂. For instance, as reported in *J. Membr. Sci.* 2021, 627, 119142, the PA membrane with MWCO of 232 Da can only produce an MgCl₂ rejection of 60%. Further characterizations are needed to clarify the chemical and pore structure of the PA membranes.

Response 4: Thanks for your comment. To be honest, we including all co-authors do not understand why the PA membrane with MWCO of 232 Da can only produce an MgCl₂ rejection of 60%. In the above literature you mentioned (*J. Membr. Sci.* 2023, 667, 121187), its retention rate for MgCl₂ is more than 55% even at the MWCO of 810 Da. In addition, the average pore radius is 0.2115 nm in this literature (*J. Membr. Sci.* 2021, 627, 119142), which is far smaller than the Stokes radius (0.347 nm) and the Hydrated radius (0.423 nm) of Mg²⁺. While its rejection of MgCl₂ is as high as 95% in literature (*Nat. Commun.* 2020, 11, 2015) when its average pore radius is 0.28 nm. In our manuscript, the average pore radius of AIP-PA membrane is 0.25 nm, and its retention of MgCl₂ is above 93%. Therefore, we think this result is totally in agreement with the literature trend.

Comment 5: The author uses MD to study the Li⁺ and Mg²⁺ transport behavior. The

dimensions of the simulation systems are about $5.54 \times 5.65 \times 28 \text{ nm}^3$ and the thickness of the PA membrane is only 4 nm in the simulated system. I think the size is too small to reflect the real membrane structure. At the very least, the authors should model the PA layer based on its thickness and pore size distribution. There is no any data to tell us the pore structure of the simulated PA membrane.

Response 5: Indeed, the size of the membrane model is smaller than the real separation membrane. However, due to the homophaneous structure, we believe, to some extent, the 4 nm-thick atomic structure of PA membrane can represent the elementary mass transfer unit in the real membrane. Moreover, for the real polymer membrane even with the μm size scale, it contains more than tens of millions atoms, at present the supercomputer cannot simulate such larger system. By the way, to our knowledge, there is almost no report about all atomic molecular dynamics simulation investigating polymer membrane separation by using the real thickness and size of the polymer membrane.

In addition, to ensure the simulation accuracy, we re-performed one simulation that $\text{Mg}^{2+}/\text{Li}^+$ is about 30 with a membrane thickness of 6 nm (please see the below **Figures R2** and **R3**). The results show that the rejection for Mg^+ and Li^+ are about 92% and 33.3%, respectively, which is consistent with our previous simulation and experimental result that the PA membrane shows an excellent Mg/Li selectivity. Therefore, we believe that the size of the membrane used in MD can reflect the real transport process in experiment, to a certain extent. More importantly, our simulations mainly discuss the properties of hydrated ions to reveal the mechanism of ions selectivity.

Figure R2 Image of recalculated simulation results

Figure R3 The recalculated simulation system

We appreciate the Review for her/his valuable comment. The details of membrane structure can be found in the “Molecular Simulation” section: “The formed maximum pore size of the PA membrane is about 0.81 nm. According to the experimental measurement (**Supplementary Fig. 6**), the polymer chain was negatively charged in this work to mimic the chemical environment inside the PA membrane ⁴⁴. The PA membrane was located at the center of the simulation box and connected to two chambers.”

Reference:

44. Pendse, A. *et al.* Highly Efficient Osmotic Energy Harvesting in Charged Boron-Nitride-Nanopore Membranes. *Adv. Funct. Mater.* **31**, 2009586 (2021).

Comment 6: The MD study does not concern the counter ion, Cl⁻. During the actual separation process, the Cl⁻ transport significantly affects the ion transport of Li⁺ and Mg²⁺, which cannot be ignored. Without the study of Cl⁻ transport, the simulated system will lead to incorrect results and conclusions.

Response 6: Actually, the counter ion (Cl⁻) has already been included in our simulation system, which can be seen in the revised **Supplementary Figure S46**. Due to the necessity of electroneutrality in the feed and permeate reservoirs, the correspondence number of counter ions (Cl⁻) also pass through the membrane to the permeate reservoirs. Herein, we mainly discuss the selectivity of Mg²⁺ and Li⁺, hence, the Cl⁻ transport is just not displayed not be ignored.

Based on the reviewer’s comments, the revised content as follow:

Supplementary Figure 46. The MD simulation system.

Comment 7: The process of transporting Li^+ and Mg^{2+} through the PA membrane is intricate, and it depends on various factors like the pore structure, surface charge, and the chemical environment of the inner pore wall. The PA membrane made from AIP has a relatively large MWCO, and its pore size distribution is not narrower than other NF membranes. This PA NF membrane cannot reject MgCl_2 beyond 90%, as far as I know. The MD simulation alone cannot fully explain why this happens, and further investigation of the chemical environment inside the PA membrane is necessary.

Response 7: Many thanks for your comments. We agree with that the transport process is intricate when ions passing through the PA membrane. For MWCO, in the literature (J. Membr. Sci. 2023, 667, 121187) you mentioned, the optional membrane (TFC0.6) has a MWCO of 810 Da (Figure S5) and a rejection rate of 55.63% for MgCl_2 (Figure 5D). In contrast, the AIP-PA membrane in our paper had a MWCO around 340 Da, and its rejection of MgCl_2 was 93.9%. I think it is consistent with its trend. Nonetheless, we still believe that MWCO does not fully and accurately reflect the pore size and distribution of nanofiltration membranes, and a comparison of MWCO alone is not appropriate. We therefore performed positron annihilation experiments and pore size distribution curve fitting (please see **Figure 2** and corresponding discussion). The positron annihilation pattern also proves that the pore size distribution within the dense layer is relatively uniform, i.e., the S-value tends to be stable in the range of 0-3 k eV. The pore size distribution curve proved that the pore size distribution of AIP-PA membrane was narrower than that of CIP-PA membrane. The pore size distribution curve proves that the pore size distribution of AIP-PA membrane is narrower than that of CIP-PA membrane. Moreover, the average pore size of AIP-PA membrane is 0.25 nm, which is smaller than the hydration radius of Mg^{2+} (0.43 nm), and thus it has the conditions to realize the retention of Mg^{2+} . In this work, in addition to discussing the effect of pore size on ion rejection, the MD simulation has also discussed the hydration shell properties of ions and the size and charge affect ions selectivity, and the results obtained all support our conclusions.

Reviewers' Comments:

Reviewer #2:

Remarks to the Author:

I propose that the authors have made a significant effort to revise the manuscript. However, I am still concerned about the MWCO of the AIP-resulted NF membrane, which is relatively larger. Using the MWCO of CIP-resulted NF membrane as a reference is insufficient to demonstrate that CIP has a superior capacity for precisely regulating the pore structure.

Responses to Reviewer 2's Comments:

Comment: I propose that the authors have made a significant effort to revise the manuscript. However, I am still concerned about the MWCO of the AIP-resulted NF membrane, which is relatively larger. Using the MWCO of CIP-resulted NF membrane as a reference is insufficient to demonstrate that CIP has a superior capacity for precisely regulating the pore structure.

Response: We appreciate the reviewer's valuable comments and high affirmation on our previous revisions. We can understand that the concerns raised by the reviewer should be "Using the MWCO of AIP-resulted NF membrane as a reference is insufficient to demonstrate that AIP has a superior capacity for precisely regulating the pore structure." Please allow us to explain in detail as follows:

On the one hand, the measured MWCO value is closely related to the molecular conformation of the series of neutral calibration substances. Usually, the measured MWCO by using chain-like PEG series is slightly larger than those determined by the near-spherical calibration substances (e.g., glycerol, glucose, sucrose, raffinose). Because the chain-like PEG series are of more permeable, this is to be expected and understandable. Even so, we believe under the same testing condition, the significant difference of MWCO between AIP and CIP illustrates the validity of the AIP strategy in terms of precisely regulating the pore structure.

On the other hand, the calculated value of MWCO is greatly affected by the retention of PEG with a lower MW. As compared to other reported PA membranes in literature^{R1-R3}, our AIP-PA membrane has a relatively lower retention of PEG-200 (equivalent Stokes radius of 3.2 Å), but has a comparable retention of PEG-400 (equivalent Stokes radius of 4.7 Å). This result therefore leads to a relatively larger value of MWCO from the fitting curve. This phenomenon also just reflects the structural uniqueness of the AIP-resulted NF membrane which can precisely separate mono- and divalent cations. The low retention of PEG-200 denotes more low-valent ions (generally hydration radius less than 3.9 Å) could permeate through the membrane, while the similar retention of PEG-400 signifies that it could remain an efficient retention of high-valent ions (generally hydration radius larger than 4.0 Å), thereby resulting in a large retention difference and a precise ionic separation.

In addition, actually there are still many examples about the reported PA membranes that have much larger MWCO than that of our AIP-PA membrane (343 Da). For example, the MWCO of crumpled PA membrane prepared on nanoparticle templates by Wang et al. is 400 Da^{R4}. The ultrathin PA membrane prepared on brush-painted single-walled carbon nanotube network support is 397 Da^{R5}. The PA membrane prepared on porous 2D MOF nanosheets by Jin et al. is 553 Da^{R6}.

Based on the reviewer's comments, we have added more detailed explanations to the revised manuscript. Please also see it as follows:

“Fig. 2j shows that the measured molecular weight cut-off (MWCO) of AIP-PA membrane is visibly smaller than that of the CIP-PA membrane. A point worth noting is that the MWCO is slightly larger than those determined by the near-spherical calibration substances (e.g., glycerol, glucose, sucrose, raffinose)²⁹⁻³¹. Taking the molecular conformation into consideration that the chain-like PEG series are of more permeable, this is to be expected and understandable. Moreover, as compared to other reported PA membranes³²⁻³⁴, our AIP-PA membrane has a relatively lower retention of PEG-200 (equivalent Stokes radius of 3.2 Å), but has a comparable retention of PEG-400 (equivalent Stokes radius of 4.7 Å). This could be another reason leading to the relatively larger value of MWCO obtained from the fitting curve. Whereas, this phenomenon just can reflect the structural uniqueness of the AIP-PA membrane that can precisely separate mono- and divalent cations. The low retention of PEG-200 denotes more low-valent ions (generally hydration radius less than 3.9 Å) could permeate through the membrane, while the similar retention of PEG-400 signifies that it could remain an efficient retention of high-valent ions (generally hydration radius larger than 4.0 Å), thereby resulting in a large retention difference and a precise ionic recognition. Nonetheless, under the same testing condition these findings further demonstrate the validity of the AIP strategy in terms of precisely regulating the pore structure^{11,35,36}. Accordingly, the calculated pore-size distribution of the former is also more sharpened. Especially, the pore size of the AIP-PA membrane is smaller (about 5.08 Å) and is exactly located between the size of monovalent and divalent ions, indicating a potent molecular sieving for selective ionic separation.”

References ([R1], [R2], [R3], [R5], [R6] as [32], [33], [34], [35], [36] have been added into the

revised manuscript)

- [R1] Xu, S. *et al.* Guanidinium manipulated interfacial polymerization for polyamide nanofiltration membranes with ultra-high permselectivity. *J. Membr. Sci.* **687**, 122003 (2023)
- [R2] Xu, X. *et al.* Anionic covalent organic framework as an interlayer to fabricate negatively charged polyamide composite nanofiltration membrane featuring ions sieving. *Chem. Eng. J.* **427**, 132009 (2022)
- [R3] Liu, L. *et al.* Modification of polyamide TFC nanofiltration membrane for improving separation and antifouling properties. *RSC Adv.* **8**, 15102-15110 (2018)
- [R4] Wang, Z. *et al.* Nanoparticle-templated nanofiltration membranes for ultrahigh performance desalination. *Nat. Commun.* **9**, 2004 (2018).
- [R5] Gao, S. *et al.* Ultrathin Polyamide Nanofiltration Membrane Fabricated on Brush-Painted Single-Walled Carbon Nanotube Network Support for Ion Sieving. *ACS Nano* **13**, 5278-5290 (2019).
- [R6] Jin, X. *et al.* Development of high permeability nanofiltration membranes through porous 2D MOF nanosheets. *Chem. Eng. J.* **471**, 14566 (2023).

Reviewers' Comments:

Reviewer #2:

Remarks to the Author:

I appreciate the authors taking the time to respond to my query. After carefully considering their explanation, I accept it.

Point-by-Point Responses to Reviewers' Comments

Reviewer #2 (Remarks to the Author):

Comment: I appreciate the authors taking the time to respond to my query. After carefully considering their explanation, I accept it.

Response: We really appreciate the reviewer's valuable suggestions, kind encouragement, and affirmation of our revised manuscript.